# Serum zinc level and hepatic fibrosis in patients with nonalcoholic fatty liver disease

**Min Chul Kim**[1◉], **Jeong In Lee**[1◉], **Jung Hee Kim**[2‡]*, **Hong Joo Kim**[1], **Yong Kyun Cho**[1], **Woo Kyu Jeon**[1], **Byung Ik Kim**[1], **Won Sohn**[1‡]*

1 Division of Gastroenterology, Department of Internal Medicine, Kangbuk Samsung Hospital, Sungkyunkwan University School of Medicine, Seoul, Republic of Korea, 2 Division of Gastroenterology, Department of Internal Medicine, Hallym University Dongtan Sacred Heart Hospital, Dongtan, Gyeonngi-do, Republic of Korea

◉ These authors contributed equally to this work.
‡ These authors also contributed equally to this work.
* wonsohn1@gmail.com (WS); mazyyang5@gmail.com (JHK)

**Data Availability Statement:** All relevant data are within the paper.

**Funding:** The author(s) received no specific funding for this work.

## Abstract

This study aimed to investigate the relationship between serum zinc level and hepatic fibrosis in patients with nonalcoholic fatty liver disease (NAFLD). A cross-sectional study was conducted using nationally representative samples from the Korea National Health and Nutrition Examination Survey 2010. Significant hepatic fibrosis was defined as Fibrosis-4 (FIB-4) index>1.3. Zinc level was measured using inductively coupled plasma mass spectrometry. Univariable and multivariable logistic regression analyses were performed to assess risk factors for significant hepatic fibrosis in patients with NAFLD. A total of 300 patients with NAFLD were analyzed in this study. The mean serum zinc level was 139.8 ±29.9 μg/dL. FIB-4 index was significantly increased as the serum zinc level decreased (Adjusted correlation coefficient = -0.177, $p = 0.003$). Significant liver fibrosis was observed in 62 patients (21%). The multivariable analysis showed that significant liver fibrosis in NAFLD was associated with diabetes mellitus (odds ratio [OR], 3.25; 95% confidence interval [CI], 1.71–6.19; $p<0.001$), male (OR, 2.59; 95% CI, 1.31–5.12; $p = 0.006$), and zinc level <140 μg/dL (OR, 2.14; 95% CI, 1.16–3.94; $p = 0.015$). There was an inverse relationship between serum zinc level and FIB-4 index in NAFLD. Low levels of serum zinc were an independent risk factor for significant hepatic fibrosis in NAFLD.

## Introduction

Nonalcoholic fatty liver disease (NAFLD) is a chronic liver disease, which is defined as intrahepatic triglyceride content of >5%. It has become widespread with the increasing prevalence of obesity and metabolic syndrome. The spectrum of NAFLD includes simple steatosis, nonalcoholic steatohepatitis (NASH), fibrosis, and cirrhosis. It also results in liver cancer [1]. NAFLD is a multifactorial disease with complex pathophysiology such as obesity, insulin resistance (IR), dyslipidemia, and metabolic syndrome [2]. Moreover, in many cases, the progression of NASH to cirrhosis remained indolent without specific symptoms until advanced liver disease.

**Competing interests:** The authors have declared that no competing interests exist.

Zinc (Zn) is an essential trace mineral element scavenging free radical oxygen species in human and plays a key factor in hepatic lipid metabolism [3]. Zn is an inhibitor of NADPH oxidase which plays an important role in the production of $O_2^{\bullet-}$ (known as reactive oxygen species from oxygen). Besides, the metallothionein, metal binding proteins produced by Zn is an excellent scavenger of $^\bullet OH$. Zn is also competing with iron and copper which catalyze the production of $^\bullet OH$ from $H_2O_2$ by binding to the cell membrane [4]. Thus, Zn acts as a potent promoter of autophagy-mediated lipophagy in the liver, reducing lipid accumulation and stimulating lipolysis [5]. In patients with chronic viral hepatitis combined with/without HIV infection, lower Zn level was not only associated with severe liver fibrosis and cirrhosis-related complications such as variceal bleeding but also increased mitochondrial oxidative stress [6, 7]. In the aspect of alcoholic liver disease, the metallothionein which is an antioxidant produced by Zn showed a cytoprotective effect on the liver [8]. Several basic studies examined the role of Zn in NAFLD or NASH. NASH or cirrhosis may alter the process of trace mineral metabolism, and decreased Zn level was reversely associated with hepatic steatosis with a leptin receptor deficiency or dysregulation of a large number of genes in lipid metabolism [9, 10]. Moreover, oxidative stress related with the change in activity of antioxidant enzyme including Zn is a well-known important pathophysiological factor in NASH progression [11].

Thus, the basic studies showed that serum Zn level and NAFLD are closely related with each other in physiologic mechanism and disease progression. Recently, the relationship between Zinc and hepatic fibrosis was revealed in patients with biopsy-proven NAFLD [12]. However, liver biopsy is not widely used as a routine screening tool to detect or monitor fibrosis progression in NAFLD because of its invasiveness and the cost. The present study aimed to examine the role of Zn in patients with NAFLD using a nationally representative database in Korea. We investigated the effect of serum Zn level on hepatic fibrosis represented by simple and noninvasive blood-based biomarker in patients with NAFLD.

## Methods

### Study population

This cross-sectional study was conducted based on data from the Korea National Health and Nutritional Examination Survey (KNHANES). The KNHANES, a nationally representative survey, is performed by the Korea Centers for Disease Control and Prevention (KCDC). It is based on a complex, stratified, multistage, and probability cluster sampling of the noninstitutionalized population in Korea [13]. This survey consists of four parts: health interview survey, nutrition survey, health behavior survey, and health examination survey. The health interview and examination were performed by trained medical staffs and interviewers at the mobile examination center. The KNHANES has been periodically performed since 1998. This study was conducted using KNHANES 2010.

The target population of this survey was all noninstitutionalized Korean civilians aged >1 year. The survey was conducted with the sampling units based on sex, age, and geographic areas, which were determined according to the household registries of the Korean National Census Registry. Written informed consent was obtained from all participants in the survey. The KNHANES was approved by the Institutional Review Board of KCDC (2010-02CON-21-C).

A total of 8,958 participants completed the survey through the mobile health examination units. First, we excluded 7,268 subjects according to the following exclusion criteria: age <19 years (n = 2,218), absence of laboratory data (n = 4,806), and absence of alcohol intake history (n = 244). Of the remaining 1690 subjects, we additionally excluded 324 patients with hepatitis B (n = 57), hepatitis C (n = 2), liver cancer (n = 2), and significant alcohol consumption

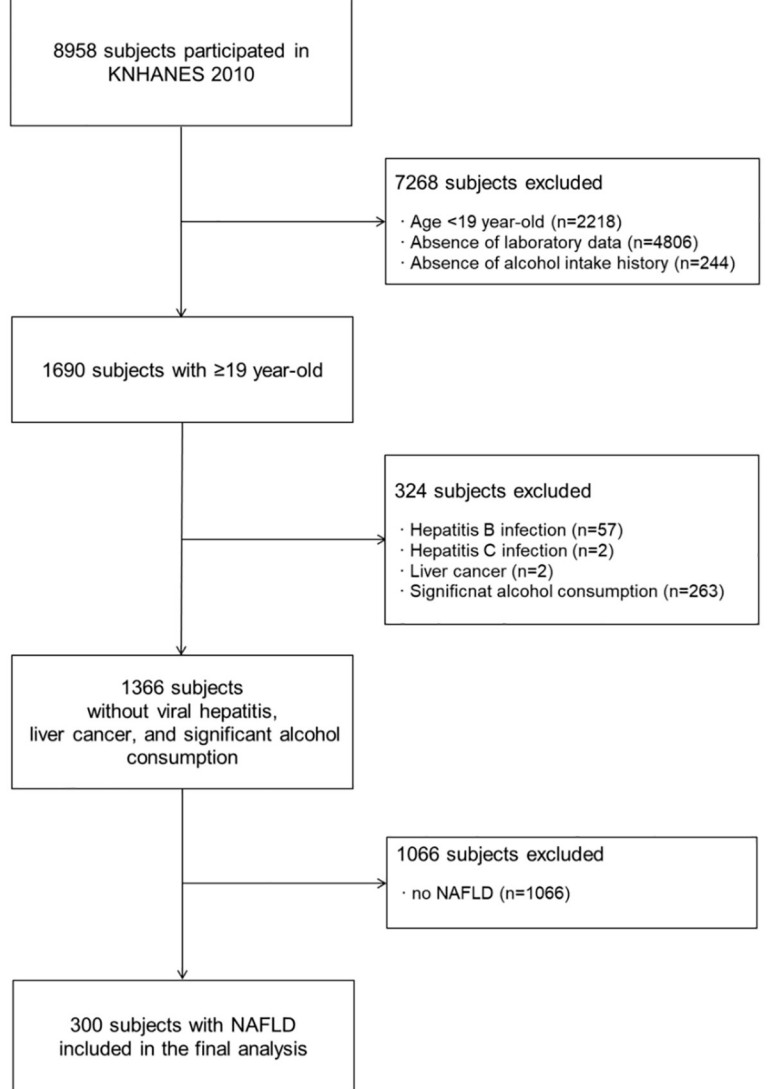

**Fig 1. Flow diagram of enrolled patients.** Abbreviations: KNHANES, Korea National Health and Nutrition Survey; NAFLD, non-alcoholic fatty liver disease.

(n = 263). Significant alcohol consumption was defined as alcohol intake >210 g/week in men or >140 g/week in women [14]. A total of 1,366 subjects with no viral hepatitis, liver cancer, and significant alcohol consumption were identified. Of 1,366 subjects, a total of 300 subjects with NAFLD were finally analyzed (Fig 1). NAFLD was defined as NAFLD liver fat score >-0.64 [15].

## Clinical variables

The information on alcohol consumption was obtained during the health interview survey. Alcohol consumption was evaluated by asking the subjects about drinking behavior during the month just before the interview. The participants were asked for their average frequency (days per month) of alcoholic beverage consumption and average amount (units of drink/day) of alcoholic beverage ingested on a single occasion. Each unit was equivalent to approximately 10

g of alcohol. Diabetes mellitus was defined as fasting blood glucose $\geq$ 126mg/dL or treated with anti-diabetic drug.

Blood tests, including aspartate aminotransferase (AST), alanine aminotransferase (ALT), gamma-glutamyl transferase (GGT), total cholesterol, low-density lipoprotein (LDL) cholesterol, triglyceride, and 25-hydroxyvitamin D (25(OH)D), were conducted after 12 h of fast. The KNHANES was conducted by the KCDC and a variety of health measurements were added to the basic design to meet emerging data needs. All health examination components including blood sample collection have been conducted in the Mobile Examination Centers (MECs) that travel to each survey location. These MECs provide a standardized environment and equipment. Routine biochemical tests, including total cholesterol, triglyceride, glucose, high-density lipoprotein (HDL) cholesterol, LDL cholesterol, ALT, and AST, were performed using the ADVIA 1650 analyzer (Bayer, Pittsburgh, PA, USA). Hepatitis B surface antigen was measured by electrochemiluminescence immunoassay method using the E-170 automated analyzer (Roche, Penzberg, Germany). Serum 25(OH)D levels were measured using a counter (1470 Wizard; Perkin-Elmer, Turku, Finland) with a radioimmunoassay method (Dia-Sorin; Still Water, MN, USA). A trace element tube was used to measure serum Zn levels. Blood samples for measuring zinc level were mixed for 10 min using a roller mixer immediately to avoid the formation of blood clots, and then were refrigerated immediately and transported in cold storage to the Central Testing Institute in Seoul, Korea. All blood samples were analyzed within 24 h after arrival at the Central Testing Institute. Zn level was measured using inductively coupled plasma mass spectrometry (ICP-MS) using PerkinElmer ICP-MS (PerkinElmer, MA, USA). IR was evaluated using Homeostatic Model Assessment of Insulin Resistance (HOMA-IR) [16]. Hepatic fibrosis was assessed using the Fibrosis-4 (FIB-4) index. Significant hepatic fibrosis was defined as FIB-4 index >1.3 [17].

## Statistical analysis

The t-test and chi-square test were used in the comparisons of continuous and categorical variables, respectively. Partial correlation analysis between serum Zn level and FIB-4 index was done adjusting potential confounding factors which could affect the relationship. Univariable and multivariable logistic regression analyses were performed to assess the risk factors, including serum Zn level, for significant hepatic fibrosis in patients with NAFLD. Multivariable analysis was done in the following manners. First, we included in the multivariable analysis any variable that was associated with the outcome at a P-value of <0.10, regardless of whether or not the variable was associated with the risk factor. Second, for multivariable logistic regression analysis, the subjects of analysis should have at least ten outcomes for each independent variable in the multivariable model. Third, multivariable analysis was performed using a forward conditional stepwise procedure to avoid multicollinearity. Thus, we included sex, BMI, LDL level, GGT level, diabetes mellitus, and zinc level in the multivariable analysis. We excluded age in the multivariable analysis because the dependent variable (FIB-4) included age. A P-value <0.05 was considered statistically significant. All statistical analyses were performed using SPSS for Window version 18.0 (SPSS Inc., Chicago, IL).

## Results

### Baseline characteristics of subjects with NAFLD

The baseline characteristics of subjects with NAFLD are presented in Table 1. The mean age of the subjects was 49.5 years. The proportion of men was 65.7% (197/300). The mean body mass index (BMI) was 26.2 kg/m$^2$. The mean AST, ALT, and GGT levels were 27.6 U/L, 33.8 U/L, and 52.0 U/L, respectively. The proportion of patients with diabetes mellitus was 21.3%

**Table 1. Baseline characteristics of the subjects with NAFLD.**

| | Total (n = 300) | Subjects with high zinc level (≥140 µg/dL) (n = 135) | Subjects with low zinc level (<140 µg/dL) (n = 165) | P-value |
|---|---|---|---|---|
| Age (year) | 49.5 ± 14.0 | 47.7 ± 13.5 | 51.0 ± 14.3 | 0.045 |
| Male | 197 (65.7%) | 98 (59.4%) | 99 (73.3%) | 0.011 |
| Height (cm) | 165.2 ± 9.0 | 166.7 ± 8.6 | 164.1 ± 9.2 | 0.012 |
| Weight (kg) | 71.8 ± 11.9 | 73.0 ± 12.3 | 70.8 ± 11.5 | 0.120 |
| BMI (kg/m$^2$) | 26.2 ± 3.3 | 26.2 ± 3.3 | 26.3 ± 3.3 | 0.872 |
| Waist circumference (cm) | 89.3 ± 8.8 | 89.8 ± 8.8 | 89.0 ± 8.8 | 0.405 |
| Platelet count (x10$^3$/mm$^2$) | 261.9 ± 68.1 | 258.8 ± 57.6 | 264.6 ± 75.7 | 0.452 |
| AST (U/L) | 27.6 ± 20.7 | 25.5 ± 10.2 | 29.2 ± 26.2 | 0.095 |
| ALT (U/L) | 33.8 ± 22.2 | 33.1 ± 17.4 | 34.4 ± 25.7 | 0.614 |
| GGT (U/L) | 52.0 ± 57.5 | 57.3 ± 60.6 | 47.7 ± 54.7 | 0.148 |
| FBS (mg/dl) | 111.3 ± 31.5 | 108.8 ± 31.0 | 113.3 ± 31.7 | 0.226 |
| Total cholesterol (mg/dl) | 198.5 ± 41.3 | 196.8 ± 37.6 | 199.9 ± 44.2 | 0.523 |
| LDL cholesterol (mg/dl) | 122.1 ± 34.5 | 120.1 ± 33.7 | 123.7 ± 35.2 | 0.261 |
| HDL cholesterol (mg/dl) | 41.6 ± 9.5 | 40.8 ± 9.8 | 42.2 ± 9.3 | 0.202 |
| Triglyceride (mg/dl) | 203.8 ± 125.4 | 217.9 ± 149.7 | 192.3 ± 100.4 | 0.079 |
| HOMA-IR score | 4.2 ± 2.8 | 4.2 ± 3.4 | 4.2 ± 3.4 | 0.941 |
| NAFLD liver fat score | 0.72 ± 1.48 | 0.68 ± 1.43 | 0.75 ± 1.51 | 0.663 |
| FIB-4 index | 0.99 ± 0.60 | 0.89 ± 0.45 | 1.07 ± 0.70 | 0.008 |
| Zinc level (µg/dL) | 139.8 ± 29.9 | 166.3 ± 21.4 | 118.2 ± 14.3 | <0.001 |
| 25(OH)D level (ng/mL) | 18.2 ± 6.4 | 19.3 ± 6.8 | 17.2 ± 5.9 | 0.006 |
| Diabetes mellitus | 64 (21.3%) | 30 (22.2%) | 34 (20.6%) | 0.734 |

*Data were represented as mean ± SD or N (%). Abbreviations: S.D., standard deviation; BMI, body mass index; AST, aspartate transaminase; ALT, alanine transaminase; GGT, gamma glutamyl transferase; FBS, fasting blood sugar; LDL, low density lipoprotein; HDL, high density lipoprotein; HOMA-IR, Homeostatic Model Assessment of Insulin Resistance; NAFLD, nonalcoholic fatty liver disease; FIB-4, fibrosis-4; 25(OH)D, 25-hydroxyvitamin D.

(64/300). The mean FIB-4 index was 0.99 ± 0.60, and the mean serum Zn level was 139.8 ± 29.9 µg/dL.

The comparison of clinical characteristics was conducted between subjects with high Zn level (≥140 µg/dL) and low Zn level (<140 µg/dL). Subjects with low Zn level were older than subjects with high Zn level (51.0 vs 47.7 years, $p = 0.045$). The proportion of men was higher in subjects with low Zn level than in subjects with high Zn level (73.3% vs 59.4%, $p = 0.011$). There were no differences in HOMA-IR value; NAFLD liver fat score; and presence of diabetes mellitus between the two groups. The serum 25(OH)D level was significantly higher in subjects with high Zn level than in subjects with low Zn level (19.3 vs 17.2 ng/mL, $p = 0.006$). There was a significant difference in FIB-4 index between the two groups. The FIB-4 index was higher in subjects with low Zn level than in subjects with high Zn level (1.07 vs 0.89, $p = 0.008$).

## Relationship between serum Zn level and hepatic fibrosis in subjects with NAFLD

The relationship between serum Zn level and hepatic fibrosis (FIB-4 index) in subjects with NAFLD are presented in Fig 2. The FIB-4 index significantly increased as the serum Zn level decreased (Pearson correlation coefficient = -0.120, $p = 0.038$). We analyzed the correlation between serum zinc and FIB-4 index adjusting potential confounding factors. The partial correlation coefficients (r) adjusted for gender, BMI, GGT, fasting glucose level, triglyceride, LDL

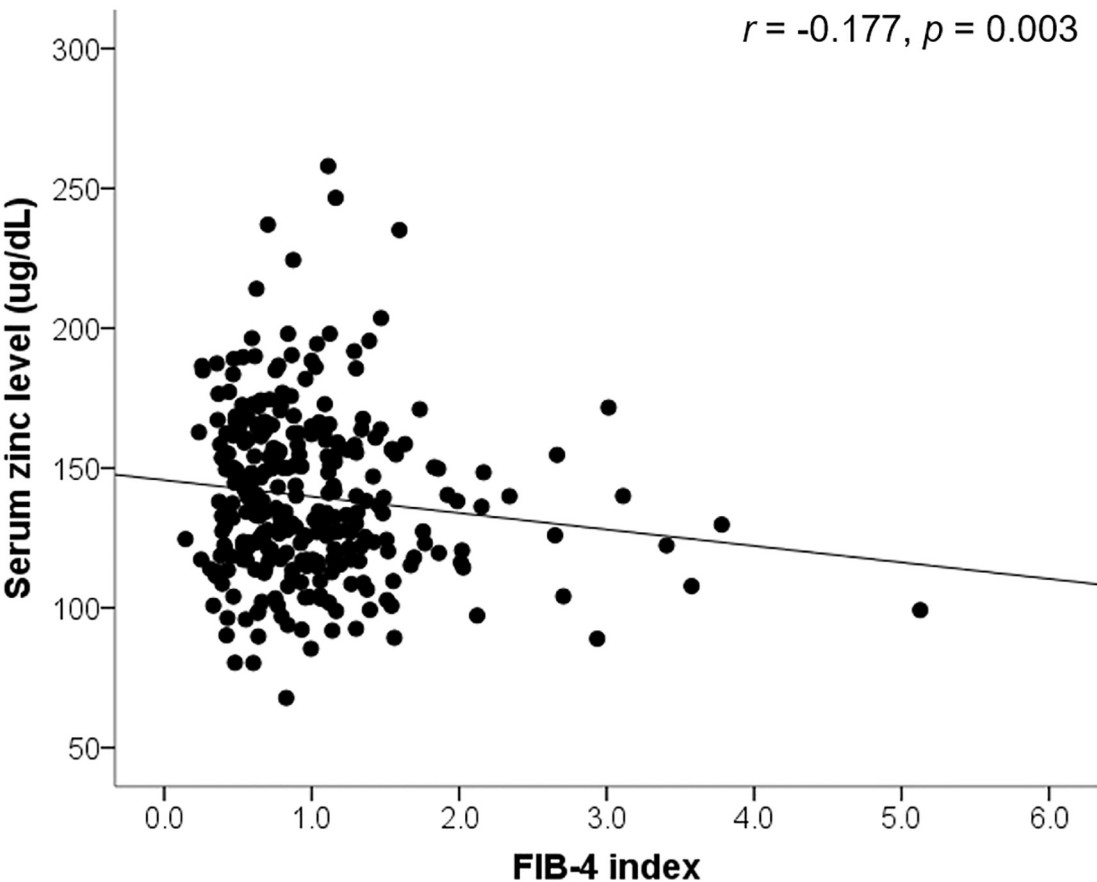

**Fig 2. The relationship between serum zinc level and hepatic fibrosis (FIB-4 index) in subjects with NAFLD.** "*r*" is Partial correlation coefficient adjusting gender, body mass index, gamma glutamyl transferase, fasting glucose level, triglyceride, low density lipoprotein cholesterol, high density lipoprotein cholesterol, diabetes mellitus, and vitamin D level. Abbreviations: NAFLD, nonalcoholic fatty liver disease; FIB-4, fibrosis-4.

cholesterol, HDL cholesterol, diabetes mellitus, and vitamin D level between serum zinc level and FIB-4 index is $r$ = -0.177 with $p$ = 0.003. We investigated the difference in FIB-4 index according to serum Zn level (<120, 120–140, 140–160, and ≥160 μg/dL) (Fig 3). The mean FIB-4 index in serum Zn level <120, 120–140, 140–160, and ≥160 μg/dL were 1.06±0.74, 1.08 ±0.66, 0.96±0.47, and 0.83±0.43, respectively. The FIB-4 index significantly increased as serum Zn level decreased ($p$ = 0.01). There is an inverse relationship between serum Zn level and hepatic fibrosis in NAFLD.

## Risk factors for significant hepatic fibrosis in subjects with NAFLD

The risk factors for significant hepatic fibrosis in subjects with NAFLD are shown in Table 2. Significant liver fibrosis (FIB-4 index ≥1.3) was observed in 62 patients (21%). The univariable analysis indicated that significant fibrosis in patients with NAFLD was associated with age ≥50 years (odds ratio [OR], 12.23; 95% confidence interval [CI], 5.07–29.49; $p$<0.001), male sex (OR, 2.05; 95% CI, 1.07–3.93; $p$ = 0.031), serum LDL level ≥130 mg/dL (OR, 0.51; 95% CI, 0.28–0.92; $p$ = 0.025), serum Zn level <140 μg/dL (OR, 1.80; 95% CI, 1.00–3.22; $p$ = 0.050), and diabetes mellitus (OR, 2.83; 95% CI, 1.53–5.24; $p$ = 0.001). The multivariable analysis was done with the forward conditional stepwise method and by six variables (sex, BMI, LDL

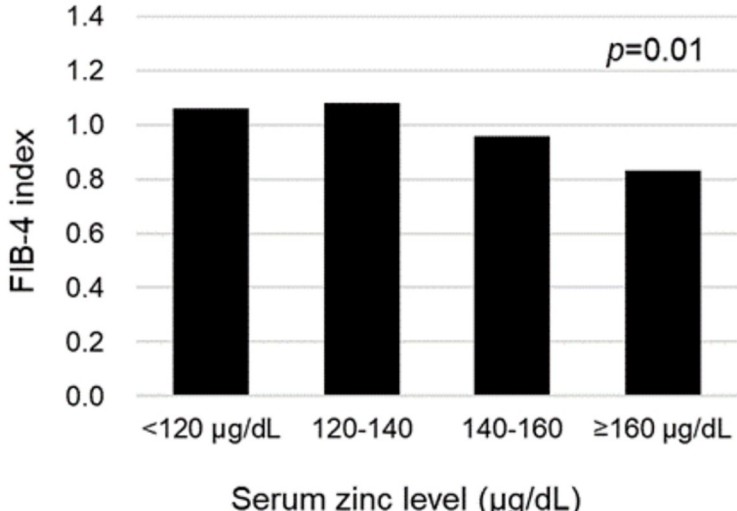

**Fig 3. Hepatic fibrosis (FIB-4 index) according to serum zinc level in NAFLD.** Abbreviations: NAFLD, nonalcoholic fatty liver disease; FIB-4, fibrosis-4.

cholesterol, GGT, diabetes mellitus, and serum zinc level) which were selected by the univariable analysis. The multivariable analysis revealed that the risk factors for significant hepatic fibrosis in subjects with NAFLD were diabetes mellitus (OR, 3.25; 95% CI, 1.71–6.19; $p<0.001$), male sex (OR, 2.59; 95% CI, 1.31–5.12; $p = 0.006$), and serum Zn level $<140$ μg/dL (OR, 2.14; 95% CI, 1.16–3.94; $p = 0.015$).

## Discussion

The present study demonstrates that there is a significant inverse relationship between serum Zn level and liver fibrosis represented by FIB-4 index in patients with NAFLD. Moreover, a serum Zn level $<140$ μg/dL is an independent risk factor for significant liver fibrosis (FIB-4 index $\geq1.3$) combined with diabetes and male sex.

Zn, a trace element of a constitution of metabolic, anti-inflammatory, and antioxidant enzymes, can have distorted metabolism in chronic liver disease [18]. Previous studies showed that patients with chronic active hepatitis, cirrhosis, and liver cancer have significantly decreased serum Zn levels [19]. Zn deficiency was shown in patients with chronic hepatitis C, and Zn level was improved by treatment with interferon [20, 21]. Thus, Zn supplementation in patients with chronic hepatitis C and liver cirrhosis improved liver function and long-term outcome, such as lower cumulative incidence of liver cancer. With the same aspect, poor survival in chronic viral patients with early HCC was significantly associated with Zn deficiency as well as incomplete infection control of the hepatitis virus [22]. The risk of esophageal variceal bleeding, the fatal complication of decompensated cirrhosis, was also correlated with the decreased level of Zn in patients with chronic viral hepatitis [6]. Thus, Zn levels are significantly associated with disease severity and prognosis in patients with chronic viral hepatitis. Zn deficiency/altered metabolism is also observed in alcoholic liver disease. Ethanol consumption leads to increased Zn excretion in the urine and decreased Zn absorption in the intestine. Chronic ethanol exposure caused a decrease in hepatic Zn level and metallothionein, the protein with substantial binding site of Zn for its function. Metallothionein containing Zn has a protective effect on alcohol-induced oxidative liver injury [23]. In a mouse model,

**Table 2. Risk factors for significant hepatic fibrosis in subjects with NAFLD.**

| | Univariable OR (95% CI) | *p*-value | Multivariable OR (95% CI) | *p*-value |
|---|---|---|---|---|
| **Age** | | | | |
| <50 years | 1 | | | |
| ≥50 years | 12.23 (5.07–29.49) | <0.001 | | |
| **Sex** | | | | |
| Female | 1 | | 1 | |
| Male | 2.05 (1.07–3.93) | 0.031 | 2.59 (1.31–5.12) | 0.006 |
| **BMI** | | | | |
| <27 kg/m$^2$ | 1 | | | |
| ≥27 kg/m$^2$ | 0.47 (0.25–0.090) | 0.023 | | |
| **Waist circumference** | | | | |
| <90 cm (male) / <85 cm (female) | 1 | | | |
| ≥90 cm (male) / ≥85 cm (female) | 1.01 (0.58–1.77) | 0.976 | | |
| **Serum LDL level** | | | | |
| <130 mg/dL | 1 | | | |
| ≥130 mg/dL | 0.51 (0.28–0.92) | 0.025 | | |
| **Serum HDL level** | | | | |
| ≥40 mg/dL (male) / ≥50 mg/dL (female) | 1 | | | |
| <40 mg/dL(male) / <50 mg/dL (female) | 0.99 (0.54–1.79) | 0.961 | | |
| **Serum triglyceride level** | | | | |
| <250 mg/dL | 1 | | | |
| ≥250 mg/dL | 0.84 (0.44–1.57) | 0.576 | | |
| **Serum ALT level** | | | | |
| <40 U/L | 1 | | | |
| ≥40 U/L | 1.06 (0.56–2.00) | 0.869 | | |
| **Serum GGT level** | | | | |
| <60 U/L | 1 | | | |
| ≥60 U/L | 1.79 (0.97–3.32) | 0.064 | | |
| **HOMA-IR level** | | | | |
| <4.2 | 1 | | | |
| ≥4.2 | 0.69 (0.37–1.29) | 0.244 | | |
| **Serum zinc level** | | | | |
| ≥140 μg/dL | 1 | | 1 | |
| <140 μg/dL | 1.80 (1.00–3.22) | 0.050 | 2.14 (1.16–3.94) | 0.015 |
| **Serum 25(OH)D level** | | | | |
| ≥20 μg/dL | 1 | | | |
| <20 μg/dL | 0.73 (0.41–1.30) | 0.284 | | |
| **Diabetes mellitus** | | | | |
| Absence | 1 | | 1 | |
| Presence | 2.83 (1.53–5.24) | 0.001 | 3.25 (1.71–6.19) | <0.001 |

*significant hepatic fibrosis was defined as FIB-4 (fibrosis-4) level was ≥1.3. Abbreviations: BMI, body mass index; ALT, alanine transaminase; GGT, gamma glutamyl transferase; FBS, fasting blood sugar; LDL, low density lipoprotein; HDL, high density lipoprotein; HOMA-IR, Homeostatic Model Assessment of Insulin Resistance; NAFLD, nonalcoholic fatty liver disease; FIB-4, fibrosis-4; 25(OH)D, 25-hydroxyvitamin D.

metallothionein overexpression displayed pathological lipid peroxidation, and protein oxidation was suppressed [8]. Moreover, Zn supplementation prevents ethanol-induced liver injury and hepatic apoptosis to inhibit oxidative stress and alcohol-induced endotoxemia. In a rat model with carbon tetrachloride-induced cirrhosis, the beneficial effect of Zn on liver cirrhosis

also was observed. Zn supplementation induced a reduction in the degree of liver injury and normalization of lipid peroxidation [24].

Previous studies were focused on the effect of Zn on viral hepatitis and alcoholic liver disease. Data on the relationship of Zn level and NAFLD are limited. In the present study, we evaluated the effect of Zn level on NAFLD, except with significant alcohol consumption based on drinking frequency and amount of alcohol intake. NAFLD has a multifactorial pathophysiology, including type 2 diabetes mellitus, dyslipidemia, and obesity. Moreover, liver is a key organ in the systemic metabolism in the development of IR and type 2 diabetes mellitus. They shared the same physiology for hepatic fat accumulation, alteration of energy metabolism, and inflammatory signals [25]. Zn is also essential in the synthesis, secretion, and storage of insulin. Thus, Zn depletion induced endoplasmic reticulum stress and cellular stress such as in chronic inflammation. Finally, it induces the development of IR and type 2 diabetes mellitus. The beneficial effects of Zn supplement on IR, glucose, and lipid profile were well reported in patients with metabolic disorders [26, 27].

Thus, we make a postulation that the alteration in Zn level was also related to liver function in patients with NAFLD, like other etiologies of chronic liver disease especially due to physiologic relationship between NAFLD and diabetes mellitus. In our study, we found that Zn level and severity of liver fibrosis (represented by FIB-4 index) had an inverse correlation, and a decreased Zn level is an independent risk factor for significant liver fibrosis in patients with NAFLD. The severity of hepatic fibrosis is a crucial risk factor in the prognosis of NAFLD in terms of the development of liver cirrhosis and liver cancer [1, 28]. Zn deficiency is common in patients with liver cirrhosis and associated with the severity of liver cirrhosis [29]. The findings of this study indicate that Zn is related to liver fibrosis in patients with NAFLD. In the rat model of NAFLD, Mousavi *et al.* showed that Zn and selenium cosupplementation reduced hepatic lipid peroxidation and angiogenesis marker [30]. Asprouli *et al.* showed that lower serum Zn levels were associated with higher severity of NAFLD assessed by ultrasonography in 189 patients while serum cesium levels showed a positive association of the severity of NAFLD [31]. Kosari *et al.* reported that serum Zn level was significantly lower in moderate and severe lobular hepatic inflammation than in mild lobular hepatic inflammation in patients with NASH although enrolled patients were younger (mean age: 36 years) than other studies and the study excluded cirrhotic patients [32]. Takanori *et al.* showed that the progression of fibrosis, but not the severity of hepatic inflammation, is associated with lower levels of serum Zn in patients with biopsy-proven NAFLD [12]. There are some differences between studies about race, geography and dietary habits which affects to amount of Zn intake. However, these studies showed a similar tendency in terms of the relationship between low Zn level and the progression of hepatic inflammation in patients with NAFLD. The present study was conducted for reliable and representative participants (KNHANES) and used the FIB-4 index which is objective and well-studied for a noninvasive estimate of liver fibrosis. These findings revealed that Zn metabolism is closely associated with the progression of chronic liver disease such as viral hepatitis, alcoholic liver disease, and even NAFLD.

This study has several limitations. First, this study did not consider the histologic findings of hepatic steatosis and fibrosis in patients with NAFLD. We assessed hepatic steatosis and fibrosis using NAFLD liver fat score and FIB-4 index. The gold standard method for the diagnosis of NAFLD and liver fibrosis is the evaluation of histological findings by liver biopsy. However, it is not widely used because of the invasiveness and the cost. This study was conducted based on a nationally representative survey complex, stratified, multistage, and probability cluster sampling of the noninstitutionalized population (the KNHANES 2010). A total of 8,958 participants completed the survey. Therefore, it is hard to check liver biopsy for the

diagnosis of NAFLD and hepatic fibrosis in these many participants. In this study, we evaluated NAFLD (NAFLD liver fat score) and hepatic fibrosis (FIB-4) using commonly used blood tests. The diagnostic accuracy of NAFLD liver fat score for NAFLD is that AUROC was 0.86–0.87 with sensitivity 86% and specificity 71% [33]. The diagnostic accuracy of the FIB-4 index for advanced fibrosis in histology is high in patients with NAFLD (mean value of area under curve, 0.80; 95% CI, 0.77–0.84) [32]. In this study, we defined the cut-off point of FIB-4 for liver fibrosis as 1.3. The reason is as follows. A meta-analysis reported that the cut-off point of FIB-4 for advanced fibrosis in NAFLD was 2.67 or 3.25 while another meta-analysis showed that the cut-off point was 1.3 [34, 35]. The cut-off point of 2.67 had an advantage in terms of positive predictive value for identifying advanced fibrosis in NAFLD while that of 1.3 had a 90% negative predictive value [36]. In the present study, the number of patients with FIB-4 >2.67 was 8 (3%) while 62 patients (21%) had the level of FIB-4 >1.3. It was difficult to perform the univariable and multivariable logistic regression analysis using the FIB-4 cut-off with 2.67. Thus, we defined the cut-off point of FIB-4 as 1.3 in this study [35]. The survey of this study did not include other noninvasive studies such as ultrasonography, transient elastography, and magnetic resonance elastography. If the study had been able to integrate non-invasive imaging techniques for the evaluation of fibrosis, the diagnostic accuracy of hepatic fibrosis would be improved because the diagnostic accuracy of FIB-4 is a little low compared to that of transient elastography or magnetic elastrography (area under the receiver operating characteristics for significant liver fibrosis: FIB-4 0.75, transient elastography 0.83, and magnetic resonance elastography 0.88) [34]. We also calculated non-invasive fibrosis markers besides FIB-4 index: (i) AST to Platelet Ratio (ARPI) score = $100 \times$ (AST / upper limit of normal) / platelet count ($\times 10^9$/L) [37]; (ii) BARD score = (BMI $\geq$ 28 = 1 point, AST to ALT ratio $\geq$ 0.8 = 2 points, diabetes = 1 point), scale 0–4 [38]. However, NAFLD fibrosis score could not be calculated because our database did not provide the information about serum albumin level. The mean level of APRI score was 0.29 while the median value of BARD score was 2. The relationship between serum zinc level and APRI score or BARD score was not stronger than that between zinc and FIB-4 index (correlation coefficients $r$ = -0.191, $p$ = 0.117 in ARPI score; $r$ = -0.120, $p$ = 0.039 in BARD score). Second, this study did not investigate the interaction between Zn and other trace elements, vitamins, and hormones, such as copper, iron, selenium, vitamin A, and testosterone. We evaluated the positive correlation between serum Zn level and vitamin D level (correlation coefficient = 0.225, $p<0.001$). However, there was no significant relationship between the FIB-4 index and serum vitamin D level (correlation coefficient = 0.077, $p$ = 0.181). Vitamin D level was not a risk factor for significant liver fibrosis in this study. Third, we were unable to determine the oral intake of Zn and serum albumin level which could affected to the lower Zn in NAFLD because the data from the KNHANES did not provide information about Zn intake and albumin. We assumed that the patients with relatively low Zn levels were more likely to have NAFLD or NASH. However, this study could not evaluate the mechanism of lower zinc level in patients with NAFLD affected by Zn intake or albumin level. Zinc-rich foods are meat, shellfish, beans, seeds, nuts, dairy foods, eggs, and whole grains. Although the KNHANES provided information about the intake of these foods, it did not provide information about oral Zn intake. Previous other studies reported that oral Zn intake was decreased in patients with NAFLD. Zolfaghari *et al.* showed that oral Zn intake in patients with NAFLD was significantly lower than that in normal healthy controls [39]. Toshimitsu *et al.* revealed that NASH patients had lower Zn intake compared to patients with fatty liver (simple steatosis) [40]. Finally, the study could not investigate the effect of Zn supplementation on the regression of liver fibrosis in NAFLD because the study was conducted based on observational studies.

## Conclusions

This study observed an inverse relationship between serum Zn level and FIB-4 index in patients with NAFLD. Low levels of serum Zn were an independent risk factor for significant hepatic fibrosis in patients with NAFLD. Zn might play a fundamental role in the pathophysiology of hepatic fibrosis in NAFLD.

## Author Contributions

**Conceptualization:** Jung Hee Kim, Yong Kyun Cho, Byung Ik Kim, Won Sohn.

**Data curation:** Min Chul Kim, Jeong In Lee, Won Sohn.

**Formal analysis:** Jung Hee Kim, Won Sohn.

**Investigation:** Hong Joo Kim, Yong Kyun Cho, Woo Kyu Jeon.

**Methodology:** Jung Hee Kim, Byung Ik Kim, Won Sohn.

**Writing – original draft:** Min Chul Kim, Jeong In Lee, Jung Hee Kim, Woo Kyu Jeon, Won Sohn.

**Writing – review & editing:** Hong Joo Kim, Yong Kyun Cho, Byung Ik Kim.

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
