## [Decision Letter · Decision Letter 0]

28 Jul 2020

PONE-D-20-13707

Serum zinc level and hepatic fibrosis in patients with nonalcoholic fatty liver disease

PLOS ONE

Dear Dr. Sohn,

Thank you for submitting your manuscript to PLOS ONE. After careful consideration, we feel that it has merit but does not fully meet PLOS ONE’s publication criteria as it currently stands. Therefore, we invite you to submit a revised version of the manuscript that addresses the points raised during the review process.

We look forward to receiving your revised manuscript.

Kind regards,

Peter Starkel, M.D., Ph.D.

Academic Editor

PLOS ONE

Journal Requirements:

2. Thank you for stating in your manuscript text "Written informed consent was obtained from all participants in the survey". Please also include this in your ethics statement.

Additional Editor Comments (if provided):

The authors must address all remarks carefully and provide the additional analyses (statititics, regression analysis etc.) requested by the reviewers. In particular, attention should be given to the interpretation of the results with regard to the cut-offs chosen and the very weak correlation with zinc levels.

Reviewers' comments:

Reviewer's Responses to Questions

**Comments to the Author**

1. Is the manuscript technically sound, and do the data support the conclusions?

Reviewer #1: No

Reviewer #2: Yes

Reviewer #3: Yes

2. Has the statistical analysis been performed appropriately and rigorously? 

Reviewer #1: No

Reviewer #2: Yes

Reviewer #3: Yes

3. Have the authors made all data underlying the findings in their manuscript fully available?

Reviewer #1: Yes

Reviewer #2: Yes

Reviewer #3: Yes

4. Is the manuscript presented in an intelligible fashion and written in standard English?

Reviewer #1: Yes

Reviewer #2: Yes

Reviewer #3: Yes

5. Review Comments to the Author

Reviewer #1: Kim MC have described the manuscript entitled “Serum zinc level and hepatic fibrosis in patients with nonalcoholic fatty liver disease”. The authors demonstrated the relationship between serum Zn level and hepatic fibrosis in NAFLD. Zn is reported to be important nutritional factor related to liver fibrosis and various types of carcinogenesis, including HCC. However, this article has some critical problems and limitations. Specific comments are given below.

1. Regarding serum Zn levels in NAFLD, recent reports have already shown the relationship between Zinc and NAFLD, and prognosis in biopsy-proven NAFLD cohort (EJGH2019, Nutrition and Cancer 2019). In this report, NAFLD participants were diagnosed with NAFLD liver fat score, not biopsy. And also, they only used Fib4 index for evaluating liver fibrosis. Liver biopsy is the gold standard for diagnosing NAFLD and assessing liver fibrosis, therefore, this study has a lack of novelty for publication.

2. The authors have set the cut off value of Fib4 index as 1.3 for evaluating significant fibrosis. As the authors have described as limitation in Discussion part, Fib4 index of 1.3 is known as the cut off point for rule out the patients with mild/no fibrosis, and the proportion of the patients with Fib4 of 2.67 or more, which is focused more for the cut off with positive predictive value, was only 8%. Fibrosis is well known as the strong factor for predicting prognosis for NAFLD patients, so the cohort in this study was not good enough, because the number of patients with significant fibrosis, which reach the level to affect the prognosis might be small, and this results less information for clinical practice.

3. How did author set the cut-off value of serum zinc level? If the authors emphasize the importance of Zn level in NAFLD, they should use the definition on Zn deficiency. Zinc deficiency is defined as serum Zn level <70 μg/dL based on previous report (Harrison’s Principles of Internal Medicine 2015). Otherwise, clinical impact should be small.

4. Following the above, mean value of serum Zn in whole participants was 139.8 μg/dL. This value is within normal limits or higher, and it is expected that the majority of participants in this study has normal liver without any hepatic inflammation and fibrosis.

5. The mechanism to induce lower Zn levels in NAFLD in this study was unclear. Hypoalbuminemia is one of the most important factors that can lead to Zn deficiency because Zn mainly binds to albumin during transportation in the blood. Additionally, hypoalbuminemia is a common metabolic disorder induced by liver cirrhosis. The authors need to add the information of serum albumin in Table1, and put albumin in logistic regression analysis.

6. Circadian variation of Zn levels has been recognized well. Were the blood samples of participants in this study collected under the same conditions, such as in the morning with a fasting state?

7. To conduct the linear regression analysis, the authors should weigh the R-values more compared with P-values. The authors showed the significant relationship in P-value (p=0.038) with Pearson correlation test, but coefficient value was -0.120. This result indicated very weak correlation on these two factors, And the statistical interpretation is not strong.

Reviewer #2: Data from a large cohort.

Limitations were partially described by the authors.

Remarks:

-only use of non invasive marker FIB4? The cutt-off used in this trial was 1.3 for significant fibrosis. Is age also taken into account?

-are there any data of echography or fibroscan in the patient group with NAFLD?

-is there any recommendation about Zinc intake of supplementation that can be given to patients with NAFLD?

Reviewer #3: In this study, the authors show that low blood zinc level is associated with higher FIB-4 score in a population of NAFLD-suspected patients (based on NAFLD liver fat score).

Major comments:

- The authors mention in the discussion that “data on NAFLD are limited” (page 12) without giving any reference. However, another study is available with the same observation based on histological results (Takanori et al. PMID 31688305), accepted one year ago. The authors should add this reference and discuss it.

- The criteria used for both NAFLD diagnosis and fibrosis are based on simple non-invasive scores. This should be clearly mentioned in the discussion. Other better non-invasive diagnostic tools exist such as transient elastography or MRI (for the evaluation of both steatosis and fibrosis).

- The Pearson correlation (between FIB-4 and Zn) is weak (-0.120). There is no figure with the graph. The authors should add it as a scatterplot, because it is the main message of the paper.

- The reason why the authors included only 6 variables in the multivariate analysis is not clear. It is true that looking for a correlation with the FIB-4 score exclude the evaluation of age, ALT, AST which are usual parameters associated with fibrosis (probably better than zinc levels). Furthermore, BMI (mentioned as included in the methods but no result available) or waist circumference should probably be included.

Minor comments:

- Numerous similar data are presented in the text and in the Table 1 (redundant information).

- The HOMA-IR is the same in the two groups (high or low zinc level). Is the HOMA-IR available in all patients?

- What is the definition of diabetes mellitus? treatment for hyperglycemia? Blood glucose > 126mg/dL?

- Was the HOMA-IR calculated/available also in treated diabetic patients? Under treatment?

- Data are available on the daily micronutrients intake in control, NAFLD and NASH patients (for example Musso et al. Hepatology 2003). Are there data on zinc intake?

- Another study with histology (Kosari et al.) is presented in the references. Histological results are also available in this study. However, in this study, there was no correlation between fibrosis and zinc levels but with portal inflammation. This should be discussed.

6. PLOS authors have the option to publish the peer review history of their article (what does this mean?). If published, this will include your full peer review and any attached files.

Reviewer #1: No

Reviewer #2: No

Reviewer #3: No

---

## [Author Response · Author response to Decision Letter 0]

21 Aug 2020

Dear Peter Starkel

We answered the reviewers’ comments point by point and revised the manuscript to carefully address all the concerns raised by the handling editor and reviewers. We attached the file 'Response to Reviewers'. We thank to the editor and the reviewers for the informative reviews and now hope that the revised manuscript is now suitable for the publication in the “Plos One”.

With best wishes. 

Yours sincerely.

Won Sohn, M.D., Ph.D.

---

## [Decision Letter · Decision Letter 1]

22 Sep 2020

Serum zinc level and hepatic fibrosis in patients with nonalcoholic fatty liver disease

PONE-D-20-13707R1

Dear Dr. Sohn,

We’re pleased to inform you that your manuscript has been judged scientifically suitable for publication and will be formally accepted for publication once it meets all outstanding technical requirements.

Kind regards,

Peter Starkel, M.D., Ph.D.

Academic Editor

PLOS ONE

Additional Editor Comments (optional):

The revised paper has been improved and all major points have been addressed.

Reviewers' comments:

Reviewer's Responses to Questions

**Comments to the Author**

1. If the authors have adequately addressed your comments raised in a previous round of review and you feel that this manuscript is now acceptable for publication, you may indicate that here to bypass the “Comments to the Author” section, enter your conflict of interest statement in the “Confidential to Editor” section, and submit your "Accept" recommendation.

Reviewer #3: All comments have been addressed

2. Is the manuscript technically sound, and do the data support the conclusions?

Reviewer #3: Yes

3. Has the statistical analysis been performed appropriately and rigorously? 

Reviewer #3: Yes

4. Have the authors made all data underlying the findings in their manuscript fully available?

Reviewer #3: Yes

5. Is the manuscript presented in an intelligible fashion and written in standard English?

Reviewer #3: Yes

6. Review Comments to the Author

Reviewer #3: The authors answered the questions well and improved their manuscript.

The limitations are clearly stated. The reference to the similar study is now mentioned.

An added sentence can be improved: page 15, line 22 " If the study could check the noninvasive imaging studies, the ..." ? This can be rewritten in this way: "If the study had been able to integrate non-invasive imaging techniques for the evaluation of fibrosis ..."

7. PLOS authors have the option to publish the peer review history of their article (what does this mean?). If published, this will include your full peer review and any attached files.

Reviewer #3: No

---

## [Editor Report · Acceptance letter]

12 Oct 2020

PONE-D-20-13707R1 

Serum zinc level and hepatic fibrosis in patients with nonalcoholic fatty liver disease 

Dear Dr. Sohn:

I'm pleased to inform you that your manuscript has been deemed suitable for publication in PLOS ONE. Congratulations! Your manuscript is now with our production department. 

Kind regards, 

on behalf of

Dr Peter Starkel 

Academic Editor

PLOS ONE